# Energy-Efficient Trajectory Planning for Smart Sensing in IoT Networks Using Quadrotor UAVs

**DOI:** 10.3390/s22228729

**Published:** 2022-11-11

**Authors:** Guoku Jia, Chengming Li, Mengtang Li

**Affiliations:** School of Intelligent Systems Engineering, Sun Yat-Sen University-Shenzhen Campus, Shenzhen 528406, China

**Keywords:** IoT, UAV-assisted communication, Dubins curve, energy-consumption model, aerodynamic effects

## Abstract

Quadrotor unmanned aerial vehicles (UAVs) are widely used as flexible and mobile access points and information carriers for the future Internet of Things (IoT). This work studies a quadrotor UAV-assisted IoT network, where the UAV helps to collect sensing data from a group of IoT users. Our goal is to optimize the UAV’s overall energy consumption required to complete the sensing task. Firstly, we propose a more accurate and mathematically tractable model to characterize the UAV’s real-time energy consumption, which accounts for the UAV’s dynamics, brushless direct current (BLDC) motor dynamics and aerodynamics. Then, we can show that the UAV’s circular flight based on the proposed energy-consumption model consumes less energy than that of hover flight. Therefore, a fly–circle–communicate (FCC) trajectory design algorithm, adopting Dubins curves for circular flight, is proposed and derived to save energy and increase flight duration. Employing the FCC strategy, the UAV moves to each IoT user and implements a circular flight in the sequence solved by the travelling-salesman-problem (TSP) algorithm. Finally, we evaluate the efficiency of the proposed algorithm in a mobile sensing network by comparing the proposed algorithm with the conventional hover-communicate (HC) algorithm in terms of energy consumption. Numerical results show that the FCC algorithm reduces energy consumption by 1–10% compared to the HC algorithm, and also improves the UAV’s flight duration and the sensing network’s service range.

## 1. Introduction

Unmanned aerial vehicles (UAVs), especially of the rotary-wing type, are widely used in in Internet of Things (IoT) communication networks due to their high mobility, autonomy and flexibility [1,2]. The concept of smart sensoring has been proposed recently, and is normally be integrated with visual and mobile monitoring systems which are assisted by UAVs, aiming to solve the challenges of conventional sensors [3]. An intelligent sensor usually consists of a sensing device, a signal conditioning, a computational block, and a communication block [4]. Furthermore, sensors with wireless communication capabilities are the key to the wireless sensor network for smart sensing [5]. For instance, in agricultural applications [6], wireless sensor nodes are distributed in farmland. UAVs help to collect the information of these nodes and send it to experts for analysis, so as to give farmers better farming suggestions. In a word, smart sensing techniques include UAVs, intelligent sensors and wireless sensor networks. Recent research on IoT communication networks can be classified in the following ways. On the one hand, according to the number of UAVs and IoT users, the communication network can be classified into three categories, namely, one-versus-one [7,8], one-versus-multi [9,10,11], and multi-versus-multi [12,13]. In [7,8], circular flight is characterized based on the derived energy-consumption model of fixed-wing UAVs. In [9], the energy consumption of all IoT users is minimized via optimizing the UAV’s trajectory in conjunction with IoT users’ wake-up schedule. In [10], the fly-and-communicate design is adopted, where the target area is swept by a UAV via a zig-zag pattern and pieces of data are broadcasted to all IoT users to minimize the UAV’s energy consumption and completion time. In [12,13], multiple UAVs are deployed to serve a set of IoT users, aiming to achieve good performance via the UAV’s trajectory. On the other hand, the trajectory design models for UAV-assisted communication can be classified into coarse and fine trajectory design. The hovering-communication (HC) design is considered as a coarse trajectory, where a UAV moves to the position directly above the IoT users and keeps hovering during communication [14]. Meanwhile, fly–hover–communication (FHC) and flying-and-communication (FAC) methods are deemed as fine trajectories, where the visited locations can be optimized and the UAV can communicate while flying, respectively [15,16]. Finally, intelligence algorithms such as deep reinforcement learning have been applied in UAV-assisted communication networks to solve optimization problems [17,18].

In spite of the advantages of line-of-sight propagation and rapid deployment, there are many problems to be solved before using UAV effectively to provide stable and reliable network support. The UAVs’ insufficient on-board battery capacity is a main challenge for wireless communication network designs. There needs to be a way to save the energy of the UAV to extend the life of network. Thus, the energy-optimized trajectory planning of UAV has become a key aspect of network management. However, an accurate energy-consumption model is required before a delicate trajectory planning. Previous works on UAV’s energy-consumption modeling can be roughly classified into three categories. The first type is power model fitting based on experimental measurements. This method requires measurements of actual power consumption under various flight conditions via an on-board power meter [19], and fitting an energy-consumption curve against user-defined variables such as flight speed [20]. It is noteworthy that the acquired energy-consumption model heavily depends on experimental environments and physical characteristics, which does not apply to all UAV models. The second type derives from the motor dynamics. Morbidi and Yacef combine the dynamic equations of brushless direct current (BLDC) motors with UAV’s dynamics to achieve the lowest energy-consumption trajectory planning [21,22,23,24]. This method is generally used in the attitude control of indoor UAV studies, whereas it does not consider the influence of aerodynamic effects, which in fact affect the energy consumption significantly. The third type derives from helicopter theory. Based on the classical helicopter theory [25,26], Zeng et al. derive an energy-consumption model for rotary-wing UAVs [15], which comprehensively considers induced power, parasitic power and blade power. Furthermore, this model has been extended to a two-dimensional model by considering the centrifugal force [27]. In addition, Yan and Yang improve this model by including acceleration [28]. Ding et al. derive a mathematical model for the quadrotor UAV in 3D flight [29]. In [30], the propulsion-energy-consumption model with deceleration and acceleration is derived for vertical flight and horizontal flight.

Nevertheless, helicopters are different from quadrotor UAVs in the following three aspects. Firstly, the blade speed of helicopters are thought to be relatively fixed, and attitude changes in helicopters are realized by adjusting the geometric pitch angle through complex mechanical structures. As a comparison, the blade pitch angle of quadrotor UAVs are fixed, and the attitude variations in quadrotor UAVs are realized by adjusting the angular velocities of four blades [31]. In addition, the main blade of helicopters are placed on the centre, while the blades of quadrotor UAVs are distributed circumferentially. Finally, as the four blades are dynamically coupled together, it is unrealistic to specify the forward speed of the quadrotor UAVs and change the speed of the four blades at the same time.

In this work, we study a quadrotor UAV-assisted communication network, where a UAV helps to collect data from a group of IoT users. Our objective is to optimize the UAV’s energy consumption while guaranteeing that IoT-user’s communication requirements are met. Firstly, we derive a trajectory-based energy-consumption model for quadrotor UAVs considering BLDC motor power as the fundamental energy-consumption source. Then, an energy-optimized trajectory algorithm is proposed by applying the energy model. The contributions are summarized as follows:1.The energy-consumption model is applicable to an arbitrary trajectory that is both physically and dynamically feasible in three-dimensional space.2.The energy-consumption model considers the four blades of quadrotor UAVs as the fundamental energy-consumption sources, rather than the single blade of helicopters.3.The energy-consumption model comprehensively takes UAV dynamics, BLDC motor dynamics, and the influence of aerodynamics into account.4.An energy-optimized, smooth, physically and dynamically feasible trajectory combined with Dubins curve is proposed in a UAV-assisted communication network.

The rest of this paper is organized as follows. Section 2 introduces the derivation of the energy-consumption model of quadrotor UAVs. In Section 3, two types of practical UAV trajectories are studied, namely, straight flight and circular flight, to characterize the UAV speed for trajectory design. In Section 4, an energy-optimized trajectory-design algorithm is proposed by applying the TSP algorithm and Dubins curve in a UAV-assisted mobile-communication network, and the last section is the summary and conclusion.

## 2. Derivation of the Energy-Consumption Model

The main idea of this energy-consumption model is to calculate the blade speed through the dynamic model of UAVs based on an input trajectory, and substitute the angular velocity into the BLDC motor equation to solve the fundamental electrical-energy consumption. Thus, the model basically contains four parts: trajectory derivation, quadrotor dynamics, aerodynamics, and BLDC motor dynamics.

### 2.1. BLDC-Motor-Dynamic Model

As quadrotor UAVs are generally driven by BLDC motors, an energy-consumption model derived from the end actuator will be more feasible and accurate. The dynamic equation for one BLDC motor is established by considering the well-known motor model shown in Figure 1.

The current across the *i*-th motor is [32]:(1)$$i_{i}\left(t\right)=\frac{1}{K_{T}}\left[m_{L}\left(\omega_{i}\right)+\left(J_{m}+J_{l}\right)\frac{d\omega_{i}}{dt}+D_{f}\omega_{i}+T_{f}\left(\omega_{i}\right)\right]$$
where $\omega_{i}$ is the angular velocity of the blade, $i_{i}$ is the current flown through the *i*-th BLDC motor, $m_{L}$ is the load torque, $K_{T}$ is the torque constant of the motor, $J_{l}$ is the load moment of inertia, $J_{m}$ is the motor moment of inertia, and $D_{f}$ is the viscous damping coefficient. $T_{f}$ is the motor friction torque, which can be ignored due to liquid lubrication [21]. Furthermore, the voltage across the *i*-th motor is:(2)$$e_{i}\left(t\right)=K_{E}\omega_{i}+Ri_{i}\left(t\right)+L\frac{di_{i}}{dt}$$
where *L* is the conductivity, *R* is the motor internal resistance, $K_{E}$ is the voltage constant, and $K_{E}=K_{T}$. Since the current is controlled fast enough by an electronic speed control (ESC), steady-state conditions can be assumed and terms with derivatives can be omitted. Thus, the propulsion power for the quadrotor UAV from one BLDC motor is:(3)$$p_{i}\left(t\right)=\frac{R}{K_{T}^{2}}{\left[D_{f}\omega_{i}+m_{L}\left(\omega_{i}\right)\right]}^{2}+\frac{K_{E}\omega_{i}}{K_{T}}\left[D_{f}\omega_{i}+m_{L}\left(\omega_{i}\right)\right]$$

Thus, the propulsion-energy consumption of the quadrotor UAV within a time duration $T_{fly}$ is:(4)$$E\left(t\right)=\int_{0}^{T_{fly}}p\left(t\right)dt$$

Note that the total propulsion power $p=\sum_{i=1}^{4}p_{i}$ will be determined when the angular velocities of all blades $\omega_{i},i=1,\cdots ,4$, are solved. In the following section, $\omega_{i}$ will be derived through UAV’s dynamic model based on an input trajectory.

### 2.2. Dynamic Model of Quadrotor UAV

The coordinate system and force analysis of a quadrotor UAV are shown in Figure 2a. An inertial reference frame {$e_{1},e_{2},e_{3}$} and a rigid body frame {$b_{1},b_{2},b_{3}$} are established. The origin of the rigid body frame is located at the center of gravity (CoG) of this drone. $b_{1}$ points in the forward direction, and the direction of $b_{3}$ is the same as that of lift. $b_{2}$ is determined by the right-hand rule. The rotation of the blades produce four torques $m_{i},i=1,\cdots ,4$ and four thrusts $f_{i},i=1,\cdots ,4$. The quadrotor UAV makes rotation and translation movements by controlling the angular velocities of the four blades. The resultant moment M and the net thrust *T* are written as:(5)$$\left[\begin{array}{c} T\\ M_{1}\\ M_{2}\\ M_{3} \end{array}\right]=\left[\begin{array}{cccc} k_{f} & k_{f} & k_{f} & k_{f}\\ 0 & -lk_{f} & 0 & lk_{f}\\ -lk_{f} & 0 & lk_{f} & 0\\ -k_{\tau } & k_{\tau } & -k_{\tau } & k_{\tau } \end{array}\right]\left[\begin{array}{c} \omega_{1}^{2}\\ \omega_{2}^{2}\\ \omega_{3}^{2}\\ \omega_{4}^{2} \end{array}\right]$$
where $k_{\tau }$ is the torque coefficient, $k_{f}$ is the thrust coefficient, *l* is the distance from the CoG of the UAV to the center of a rotor shaft, $M_{1}$, $M_{2}$ and $M_{3}$ are the moments around axis $b_{1}$, $b_{2}$ and $b_{3}$, respectively. $T=\sum_{i=1}^{4}f_{i},M={\left[M_{1},M_{2},M_{3}\right]}^{T}$.

The dynamic equations are established as shown in Figure 2b:(6)$$m\ddot{q}=TRe_{3}-mge_{3}-D\frac{\dot{q}}{\dot{∥q∥}}$$
(7)$$J\dot{\Omega }=M-\Omega \times J\Omega$$
(8)$$\dot{R}=R\hat{\Omega }$$
(9)$$D=\frac{1}{2}\rho S_{F}{∥\dot{q}∥}^{2}$$
where $\Omega \in R^{3}$ is the angular velocity of the vehicle in rigid body frame, *R* is the rotation matrix from rigid body frame to inertial reference frame, *J* is the inertia matrix $J=diag[I_{x},I_{y},I_{z}]$, q is the input trajectory in frame {e}, *m* is the mass of the rigid body, *g* is the gravitational acceleration, *D* is the fuselage drag, $S_{F}$ is the fuselage equivalent flat plate area, and ρ is the air density. The operator ∧ above Ω is defined as follows: for any $x,y\in R^{3}$, there is $\hat{x}y=x\times y$. The operation ∨ is the opposite of the operation ∧.

The dynamic model of UAV in this part can be extended to multi-rotor UAVs by adding the extra angular velocities from additional rotor blades in (Equation 5). The number of angular velocities is equal to the number of rotors. The additional angular velocity will not affect the subsequent process, as only resultant moment M and net thrust *T* are required.

### 2.3. Trajectory and Force

In this part, we will find a method to solve the desired forces and moments based on the UAV’s desired trajectory. Given a desired trajectory $q_{d}$, the thrust $T_{d}$ required to track the trajectory can be obtained directly by (Equation 6) and (Equation 9):(10)$$T_{d}=m{\ddot{q}}_{d}+mge_{3}+\frac{1}{2}\rho S_{F}∥{\dot{q}}_{d}∥{\dot{q}}_{d}$$

The first term on the right-hand side of (Equation 10) means that an accacceleration is generated by the resultant external force acting on the UAV. The second term is the gravitational force in the opposite direction of $e_{3}$. The third term is the fuselage drag, which is opposite to the forward direction. Note that only $q_{d}$ is variable on the right side of the equation, which means $T_{d}$ can be totally determined by $q_{d}$. In addition, the desired net thrust $T_{d}$ here is a vector symbol, which is different from the scalar sign *T* in the previous part.

The desired resultant torque $M_{d}$ for a UAV to track the trajectory can be obtained by the transformation of (Equation 7):(11)$$M_{d}=J{\dot{\Omega }}_{d}+\Omega_{d}\times J\Omega_{d}$$

The solution for $M_{d}$ is a little bit indirect but can be obtained through a series of derivations:(12)$$\Omega_{d}={\left(R_{d}^{T}{\dot{R}}_{d}\right)}^{\vee }$$

Take the derivative of Equation (Equation 12):(13)$${\dot{\Omega }}_{d}={\left({\dot{R}}_{d}^{T}{\dot{R}}_{d}+R_{d}^{T}{\ddot{R}}_{d}\right)}^{\vee }$$

Now the problem in (Equation 11) turns to solving $R_{d}$, ${\dot{R}}_{d}$ and ${\ddot{R}}_{d}$ to obtain $\Omega_{d}$, ${\dot{\Omega }}_{d}$ and eventually $M_{d}$. In the following part, the expression of $R_{d}$, ${\dot{R}}_{d}$ and ${\ddot{R}}_{d}$ will be formulated by $q_{d}$ and $T_{d}$. More details can be found in [33].

#### 2.3.1. The Expression of $R_{d}$

The three column vectors of $R_{d}$ correspond to the three direction axes of the rigid body frame, respectively. Firstly, the direction of thrust is, by definition, the same as that of $b_{3d}$ in frame {b}, $b_{3d}=\frac{T_{d}}{∥T_{d}∥}$. Secondly, the direction of $b_{1d}$ is the same as that of velocity, $b_{1d}=\frac{{\dot{q}}_{d}}{∥{\dot{q}}_{d}∥}$. Finally, the expression of $b_{2d}$ is determined by the right-hand rule, $b_{2d}=b_{3d}\times b_{1d}$. Thus, the rotation matrix $R_{d}$ is written as:(14)$$R_{d}=\left[b_{1d},b_{2d},b_{3d}\right]$$

#### 2.3.2. The Expression of $\dot{R_{d}}$

${\dot{b}}_{3d}$ can be obtained by taking the derivative of $b_{3d}$ with respect to time:(15)$${\dot{b}}_{3d}=\frac{{\dot{T}}_{d}{∥T_{d}∥}^{2}-T_{d}\left(T_{d}^{T}{\dot{T}}_{d}\right)}{{∥T_{d}∥}^{3}}$$

As the $b_{1d}$ axis is the same as the velocity direction, ${\dot{b}}_{1d}$ is the same as the acceleration direction, ${\dot{b}}_{1d}=\frac{{\ddot{q}}_{d}}{∥{\ddot{q}}_{d}∥}$. The expression of ${\dot{b}}_{2d}$ can be obtained by the derivation of $b_{2d}$ with respect to time, ${\dot{b}}_{2d}={\dot{b}}_{3d}\times b_{1d}+b_{3d}\times {\dot{b}}_{1d}$. Thus, rotary speed matrix ${\dot{R}}_{d}$ is written as:(16)$${\dot{R}}_{d}=\left[{\dot{b}}_{1d},{\dot{b}}_{2d},{\dot{b}}_{3d}\right]$$

#### 2.3.3. The Expression of ${\ddot{R}}_{d}$

Take the derivative of ${\dot{b}}_{3d}$ with respect to time:(17)$$\begin{array}{cc} {\ddot{b}}_{3d}= & \frac{{\ddot{T}}_{d}}{{∥T_{d}∥}^{2}}-\frac{{\dot{T}}_{d}\left(2T_{d}^{T}{\dot{T}}_{d}\right)}{{∥T_{d}∥}^{3}}+\frac{3T_{d}{\left(T_{d}^{T}{\dot{T}}_{d}\right)}^{2}}{{∥T_{d}∥}^{5}}-\frac{T_{d}\left({\dot{T}}_{d}^{T}{\dot{T}}_{d}+T_{d}^{T}{\ddot{T}}_{d}\right)}{{∥T_{d}∥}^{3}} \end{array}$$

Similarly, ${\ddot{b}}_{1d}$ is the jerk direction, ${\ddot{b}}_{1d}=\frac{{\dddot{q}}_{d}}{∥{\dddot{q}}_{d}∥}$. Finally, ${\ddot{b}}_{2d}={\ddot{b}}_{3d}\times b_{1d}+2{\dot{b}}_{3d}\times {\dot{b}}_{1d}+b_{3d}\times {\ddot{b}}_{1d}$. Hence, the rotary acceleration matrix ${\ddot{R}}_{d}$ is written as:(18)$${\ddot{R}}_{d}=\left[{\ddot{b}}_{1d},{\ddot{b}}_{2d},{\ddot{b}}_{3d}\right]$$

When $\Omega_{d}$ and ${\dot{\Omega }}_{d}$ are solved by substituting $R_{d}$, ${\dot{R}}_{d}$ and ${\ddot{R}}_{d}$ into (Equation 12) and (Equation 13), we ultimately obtain $M_{d}$ as in (Equation 11) based on the input trajectory $q_{d}$. Note that the trajectory is required to be third-order derivable.

### 2.4. Thrust Coefficient and Torque Coefficient

Since an accurate model is not necessarily needed for UAV-control algorithm studies, the torque coefficient and thrust coefficient of a quadrotor UAV are usually assumed to be constants [34,35]. Nevertheless, these two coefficients are affected by the flight state, furtherly affecting the energy consumption of a quadrotor UAV [36,37]. Due to the structural characteristics of quadrotor UAVs, the thrust coefficient and torque coefficient in vertical and horizontal directions are different. Consequently, the energy-consumption calculation should be divided into horizontal-flight and vertical-flight calculations. The influence of aerodynamics on torque coefficient and thrust coefficient for axial flight and forward flight are analyzed in the Appendix A.

### 2.5. Energy-Consumption Calculation Algorithm

Combining the contents of the previous four sections, the propulsion power and energy consumption of a quadrotor UAV can be solved based on a desired trajectory $q_{d}$. As mentioned in the BLDC-motor dynamic section, the angular velocity of the blades $\omega_{i}$ should be acquired firstly. This can be carried out by substituting coefficients (Equation 43) and (Equation 44), desired torque $M_{d}\left(q_{d}\right)$ and desired force $T_{d}\left(q_{d}\right)$ in the trajectory and force section into (5) in the UAV dynamic model section. Then, (5) can be rewritten as:(19)$$\sum_{i=1}^{4}\left(C_{1}\omega {}_{i}^{2}+C_{2}\omega_{i}+C_{3}\right)=T_{d}\left(q_{d}\right)$$
(20)$$lC_{1}\left(\omega {}_{4}^{2}-\omega {}_{2}^{2}\right)+lC_{2}\left(\omega_{4}-\omega_{2}\right)=M_{1d}\left(q_{d}\right)$$
(21)$$lC_{1}\left(\omega {}_{3}^{2}-\omega {}_{1}^{2}\right)+lC_{2}\left(\omega_{3}-\omega_{1}\right)=M_{2d}\left(q_{d}\right)$$
(22)$$\sum_{i=1}^{4}{\left(-1\right)}^{i}\left(D_{1}\omega {}_{i}^{2}+D_{2}\omega_{i}+D_{3}\frac{1}{\omega_{i}}\right)=M_{3d}\left(q_{d}\right)$$
where $C_{1}=\frac{1}{6}\rho sAa\theta_{0}r^{2}$, $C_{2}=-\frac{1}{4}\rho sAav_{i0}r$, $C_{3}=\frac{1}{2}\rho sAa\theta_{0}V^{2}$, $D_{1}=\frac{1}{8}\delta \rho sAar^{3}$, $D_{2}=\frac{1}{6}\left(1+k\right)v_{i0}\rho sA\theta_{0}r^{2}$, $D_{3}=\frac{1}{2}\left(1+k\right)v_{i0}\rho sAa\theta_{0}V^{2}+\frac{1}{8}\rho S_{F}V^{3}$. The coefficients $C_{1},C_{2},C_{3},D_{1},D_{2},D_{3}$ are determined by the specified forward speed *V* based on an input trajectory $q_{d}$. Thus, the angular velocity of the blades $\omega_{i}$ can be obtained by (Equation 19)–(Equation 22). Then, the values of $\omega_{i}$ are substituted back into (Equation 3) and (Equation 4) in the BLDC motor dynamic seciton to calculate the propulsion power and energy consumption of a quadrotor UAV. Note that the degree of the equation in (Equation 20) is more than 5, thus leading to no analytical solution. However, (Equation 19)–(Equation 22) can be numerically solved quite easily by modern computers. The energy-consumption calculation algorithm is concluded in Algorithm 1.
**Algorithm 1** Trajectory-based quadrotor UAV’s energy-consumption calculation. 1: **Input**: $q_{d}$ and the parameters listed in Table 1. 2: Solve for $T_{d}$ and $M_{d}$ using (10) and (11). 3: Solve for $t_{c}$ and $q_{c}$ using (30)(31) or (32)(35). 4: Substitute $T_{d}$, $M_{d}$, $t_{c}$ and $q_{c}$ into (38) to obtain $\omega_{i}$. 5: Substitute $\omega_{i}$ into (3) and (4) to obtain *p* and *E*. 6: **Output**: *p* and *E*.


## 3. Trajectory Study

In this section, simulation examples are established to prove the feasibility and validity of the proposed energy-consumption model. As mentioned earlier, the thrust coefficient and torque coefficient in vertical and horizontal directions are normally distinct. Consequently, the flight simulation is limited to ascending, descending and any trajectory in the horizontal plane, which is adequate for actual application scenarios. A conventional quadrotor UAV’s physical parameters are listed in Table 1.

### 3.1. Linear Trajectory

Considering a linear trajectory with constant speed $q_{d}\left(t\right)={\left[Vt,0,0\right]}^{T}$, the propulsion power of a quadrotor UAV is calculated as a function of forward speed *V* and plotted with the blue line in Figure 3. With the increase in forward speed, the propulsion power decreases firstly and then increases, which is consistent with the propulsion-power curve of helicopters [25]. The reason behind this is that the torque coefficient and thrust coefficient are affected by the forward speed. As a result, the power consumption of a quadrotor UAV moving forward at low speed is lower than that when hovering. What is more, the speed that maximizes to the total travel distance for a fixed amount of energy is denoted as $V_{minE}$, which can be acquired graphically by taking the tangential line from the origin to the power curve [15]. Note that the UAV speed that minimizes the propulsion power, denoted as $V_{minp}$, is not the optimal speed to achieve the minimum energy consumption, as it needs a longer time to travel a fixed distance, compared with $V_{minE}$.

In contrast, the propulsion power for a rotary-wing UAV in [15] is:(23)$$\begin{array}{cc} P(V,\tau )= & P_{0}\left(1+\frac{3V^{2}}{\omega^{2}r^{2}}\right)+P_{i}\tau {\left(\sqrt{\tau^{2}+\frac{V^{4}}{4v_{0}^{4}}}-\frac{V^{2}}{2v_{0}^{2}}\right)}^{1/2}+\frac{1}{2}d_{0}\rho sAV^{3} \end{array}$$
where $\tau =\frac{T}{mg}$, $P_{0}=\frac{\delta }{8}\rho sA\omega^{3}r^{3}$ and $P_{i}=\left(1+k\right)\frac{{\left(mg\right)}^{3/2}}{\sqrt{2\rho A}}$. The blade angular velocity ω in [15] is deemed as a constant, resulting that $P_{0}$ and $P_{i}$ become two constants. Meanwhile, the angular velocities of the four blades are definitely different for the quadrotor UAVs. For ease of comparison, the simulation parameter of ω is chosen as 600 rad/s, which is the same as the averaged value calculated by the proposed method in this paper, and all other simulation parameters are the same as those in Table 1. Then, the results calculated by the method in [15] are shown with the red line in Figure 3. Note that the value is multiplied four times for the sake of comparison, as the model in this letter has four times more blades. This is also consistent with the fact that the stability of multiple-rotor UAVs is achieved by lower energy efficiency. During rotor rotation, four torques are formed in the opposite direction to rotation. In order to overcome the influence of torques, two of the four rotors rotate clockwise and two anticlockwise. Counter torques counteract each other, which will consume additional energy. For instance, the hexarotor UAV is more stable than a quadrotor UAV, but it consumes more energy under the same physical parameters. In addition, the model in [15] is derived from the energy consumed by the blade and fuselage without considering the existence of the efficiency of motor. It can be concluded that the configuration of a rotary-wing UAV has an important impact on energy consumption. To be specific, the more rotors, the more energy consumption of a rotary-wing UAV under the same parameters.

### 3.2. Circular Trajectory

Considering a circular trajectory in the horizontal plane $q_{d}\left(t\right)={\left[cos\left(\pi t\right),sin\left(\pi t\right),0\right]}^{T}$, t∈[0,10] s, an intuitive simulation display in MATLAB is depicted in Figure 4a. The calculated energy consumption of four motors and an individual motor are shown in Figure 4b.

The results of simulations indicate that the energy consumption of a UAV increases linearly with time, as the propulsion power of a UAV tracking circular trajectory is constant, even if it is not a steady-state motion. Note that the energy consumptions of the four motors are different, meaning that the angular velocities of the four blades are not exactly the same due to the existence of linear and angular accelerations. It is worth noting that the energy consumption of a circular trajectory is greater than that of a linear trajectory at the same forward speed. As the UAV needs to keep changing its heading direction, is requires additional energy. In addition, the energy consumption of a circular trajectory is also related to the circular radius. The smaller the radius, the greater the energy consumption. As trajectories with a smaller radius have a larger centrifugal force, they require more severe direction changing. On the contrary, a larger radius means lower propulsion power. When the radius tends to infinity, the circular trajectory becomes a linear trajectory. Note that the propulsion power of a linear trajectory at low speed forward is lower than that of hovering. Integrated by time, thus, a circular trajectory at low speed for UAV-assisted communication can also achieve less energy consumption than hovering, which will be applied to the UAV-assisted communication networks in the next section.

## 4. Energy-Optimized Trajectory Planning for Mobile-Communication Network

In this section, we will firstly introduce the communication model, and then compare the total energy consumption of a UAV collecting data from one IoT user via circular and linear trajectories to characterise them, and, finally, propose an energy-saving, smooth, physically and dynamically feasible trajectory combined with a Dubins curve.

### 4.1. System Model

A UAV-assisted wireless communication network is considered, in which a quadrotor UAV is employed to collect data from a group of *K* IoT users, as is shown in Figure 5.

It is assumed that the quadrotor UAV flies horizontally at a fixed altitude *H*, which could be the proper altitude required for obstacle avoidance without descending or ascending frequently. The quadrotor UAV’s position projected onto the horizontal plane is denoted by $q\left(t\right)={\left[x_{uav}\left(t\right),y_{uav}\left(t\right)\right]}^{T}\in R^{2}$. The location of the *k*-th IoT user is denoted as $u_{k}\in R^{2}$. Consequently, the distance between the UAV and the *k*-th IoT user is written as:(24)$$d\left(t\right)=\sqrt{H^{2}+{∥q\left(t\right)-u_{k}∥}^{2}}$$

It is assumed that the communication link between the IoT users and the UAV is the line-of-sight (LoS) channel. In addition, it is assumed that the Doppler effect caused by UAV movement is well-compensated. Hence, the channel capacity throughout in bits/second based upon the free-space path loss model can be established as:(25)$$R\left(t\right)=B\log_{2}\left(1+\frac{\gamma_{0}}{H^{2}+{∥q\left(t\right)-u_{k}∥}^{2}}\right)$$
where *B* represents the communication bandwidth, $\gamma_{0}=p\beta_{0}/\sigma^{2}$ is defined as the received signal-to-noise ratio (SNR), $\beta_{0}$ represents the power gain when the reference distance is 1 m, *p* denotes the transmitted power at the transmitter, and $\sigma^{2}$ denotes the noise power. During a communication period *T*, the total amount of data transmitted is:(26)$$Q=\int_{0}^{T}R\left(t\right)dt$$

For circular and hover flight approaches, the distances between the UAV and IoT users are both constants once the radius and altitude are determined. Thus, the communication time *T* is conversely determined by the amount of data and the communication rate R(t).

### 4.2. Comparison between Circular and Hover Flight

In this part, two types of UAV trajectories which are simple but practical are applied in the UAV-assisted wireless communication network. As is shown in Figure 6, the first one is the circular flight where a UAV flies around an IoT user following a circular trajectory. The second one is the hover flight where a UAV is directly hovering above an IoT user.

In the previous section, the power and energy consumption of a linear trajectory and circular trajectory are calculated. However, the energy consumption is also related to the completion time in a UAV-assisted communication network. For ease of comparison, the energy consumption of the circular flight and hover flight will be calculated separately. Note that the communication-related energy is ignored because it is too small compared to the propulsion energy, e.g., a few watts [38] versus hundreds of watts [20].

The energy calculation for hovering communication is relatively simple. Firstly, the distance between UAV and IoT users in (Equation 24) becomes constant, that is, exactly equal to the altitude *H* in Figure 6. Thus, the channel capacity throughout $R_{hov}$ also becomes constant. In addition, the hovering power $P_{hov}$ is determined by the physical parameters of a UAV. Hence, the energy consumption of a hovering UAV can be calculated by:(27)$$E_{hov}=\frac{Q_{data}}{R_{hov}}P_{hov}$$

As for the circular flight, the distance in (Equation 24) becomes constant for a given radius $r_{cir}$, which is $d=\sqrt{H^{2}+r_{cir}^{2}}$. Therefore, the channel capacity throughout is determined by the radius of circle, denoted as $R(r_{cir})$. With the physical parameters of a UAV as listed in Table 1, the propulsion power of a circular flight, denoted as $P_{cir}$, becomes a function of the flight radius and flight speed. Thus, the energy consumption of a UAV for circular flight can be expressed as:(28)$$E_{cir}=\frac{Q_{data}}{R(r_{cir})}P_{cir}\left(r_{cir},v_{cir}\right)$$

With the communication parameters of Table 2, the propulsion energy consumption of a quadrotor UAV in circular flight can be calculated as a function of flight radius and flight velocity, as shown in Figure 7. The optimal speed and circular radius for minimizing energy consumption are denoted as ($v_{cir}^{*},r_{cir}^{*}$). Numerical calculation in MATLAB shows that the value of $v_{cir}^{*}$ is 8.1 m/s, and the value of $r_{cir}^{*}$ is 33.1 m, leading to the lowest value of energy consumption $E_{cir}^{*}$(148.68 kJ). We find that the energy consumption of hovering flight $E_{hov}$ is 155.16 kJ, namely, $E_{cir}^{*}<E_{hov}$. By selecting different communication parameters, it is found that the amount of communication data does not affect the value of ($v_{cir}^{*},r_{cir}^{*}$). On the contrary, the altitude of a UAV has an important influence on the optimal value. Therefore, the circular flight can be used to replace hover flight in UAV-assisted communication networks under appropriate conditions.

### 4.3. Trajectory Planning

Our goal is to design a simple but practical trajectory for a UAV to optimize the energy consumption of the total distance required to accomplish the communication task with all IoT users. To accomplish the task described in Figure 5, we firstly take into account the HC protocol. Namely, the UAV, in sequence, visits a group of hovering locations which are directly above the IoT users, and communicates with each of the IoT users only at the corresponding hovering locations. The problem is reduced to seeking out the optimal visiting order $\left\{{\hat{\pi }}_{k}\right\}$ and the UAV velocities of the straight-level flight $V_{minE}$. The visiting order problem is equivalently a travelling salesman problem (TSP), and can be approximately solved by leveraging numerous existing algorithms [39]. The optimal UAV speed $V_{minE}$ can be acquired graphically by taking the tangential line from the coordinate origin to the power curve in Figure 3, as mentioned in the previous section. In [14], the HC design is considered as a coarse trajectory design. Based on this terminology, we propose a fine trajectory which is named the fly-circle-communicate (FCC) design, by replacing hover flight with a circular flight in order to achieve lower energy consumption. In addition, the entry and exist positions of circular trajectory are connected by a Dubins curve to realize a smooth transition without severe acceleration or deceleration.

Dubins-curve path planning aims at seeking out the shortest smooth path between two points which are associated with a determined orientation angle and bounded curvature [40]. The research result summarized by Dubins gives a sufficient set of paths comprised of three path segments which are either circular-arcs or straight-line segments. The sequence is denoted as CCC or CSC, where *S* denotes a straight-line segment, and *C* represents a circular arc. Each *C* has two choices, namely, turning right or turning left, which are denoted as *R* and *L*, respectively. Hence, the Dubins set $D_{path}$ consists of six allowed paths, $D_{path}=\left\{LSL,LSR,RSL,RSR,RLR,LRL\right\}$.

The trajectory proposed in this work has two stages. In the first phase, the UAV flies in a straight line to a position near to one IoT user at the optimal speed $V_{minE}$. In the second stage, the UAV flies around in a circular trajectory while collecting data from IoT users. Meanwhile, the communication time is determined by the data amount of IoT users. The radius of the circle is chosen to be the same value as $r_{cir}^{*}$, and the velocity along the circluar path is chosen as $v_{cir}^{*}$.

The entry position of the arc, denoted as $p_{1}$, is the point of tangency of the next circle. The position calculation of the point of tangency is analyzed below, as shown in Figure 8.

We assume that the circular trajectories are all counterclockwise. The positions of the IoT users have been specified, so the center distance *s* between the two circles is known. Thus, the distance between the center $O_{1}$ and the point of tangency $p_{1}$ is:(29)$$l=\sqrt{s^{2}-{r_{cir}^{*}}^{2}}$$α is the orientation of $p_{1}$:(30)$$\alpha =\arctan\frac{O_{2}\left(y\right)-O_{1}\left(y\right)}{O_{2}\left(x\right)-O_{1}\left(x\right)}-\theta$$
where θ can be calculated:(31)$$\theta =\arctan\frac{r_{cir}^{*}}{l}$$

Thus, the position of $p_{1}$ is:
(32a)$$\begin{array}{c} y=l\sin\alpha +O_{1}\left(y\right) \end{array}$$
(32b)$$\begin{array}{c} x=l\cos\alpha +O_{1}\left(x\right) \end{array}$$

The exit position $p_{2}$ is determined by the entry position of the arc and the circle track left by the UAV during communication. Finally, the final trajectory will be formed by connecting the exit position of the first circle and the entry position of the next circle segment by segment using Dubins curves. The trajectory-design algorithm is concluded as follows (Algorithm 2).
**Algorithm 2** Energy-optimized trajectory planning algorithm. 1: **Input**: IoT user locations $\left\{u_{k}\right\}$, data amount $Q_{k}$, and the parameters listed in Table 1 and Table 2. 2: Apply TSP algorithm to acquire the optimal visiting order $\left\{{\hat{\pi }}_{k}\right\}$. 3: Solve $v_{cir}^{*}$ and $r_{cir}^{*}$ numerically for the circular trejectory, and choose $V_{minE}$ for the linear trajectory. 4: **for** m=1:k 5:     Solve $p_{1}$ and $p_{2}$ of the circular trajectory centered on IoT user(${\hat{\pi }}_{m}$) using (45)–(48). 6:     Connect $p_{2}$ of IoT user(${\hat{\pi }}_{m}$) and $p_{1}$ of IoT user(${\hat{\pi }}_{m+1}$) by applying Dubins-curve algorithm. 7: **end for** 8: **Output**: energy-optimized UAV trajectory.


Note that the proposed trajectory-planning algorithm is suitable for sparse IoT user distribution. To be specific, the distance between two IoT users should be at least two times larger than the radius of the circle to highlight the merits of the circular trajectory.

### 4.4. Application Studies

In this part, the proposed FCC trajectory designs are numerically compared with HC designs for energy consumption, as it is hard to mathematically prove that FCC consumes less energy than HC. With the parameters of the communication model in Table 2, the FCC trajectories and HC trajectories are illustrated in Figure 9.

We randomly initialized the positions of 10 IoT users in Figure 9a and 15 IoT users in Figure 9b, and implemented the proposed trajectory to command a UAV to collect data. The base station location of a UAV is [0,0], which should be regarded as one IoT user in the TSP algorithm. The UAV starts from the base station and returns to the base station after completing the task. The dots in the center of the red circle represent the positions of IoT users. HC trajectory is denoted as the black dotted line, which is formed by connecting the positions of IoT users in the order solved by the TSP algorithm. The UAV will collect data from IoT users only when it is directly above the IoT users at the altitude of *H*. As a comparison, the FCC trajectory is drawn with a blue solid line and red solid circular line, which is formed by applying Algorithm 2. The UAV firstly moves to each of the IoT users in sequence in the direction indicated by the arrow. Then, the UAV will implement the circular flight and collect data from each of the IoT users simultaneously when the UAV reaches the boundary of the circular trajectory.

In Figure 10, the propulsion power and energy consumption of a quadrotor UAV applying HC and FCC methods are calculated as a function of time. The positions of IoT users are the same as that in Figure 9a. To be specific, the horizontal ordinates of 10 IoT users are $u_{x}={\left[42,-745,-371,-530,1175,-307,374,-96,444,-382\right]}^{T}$ m, and the longitudinal coordinates of 10 IoT users are $u_{y}={\left[-701,-711,244,-88,-98,709,145,99,794,-402\right]}^{T}$ m. The data amount of IoT users are also randomly generated, which are $Q_{data}={\left[1.2,1,0.1,0.7,0.4,1.6,0.7,1.1,0.4,1.3\right]}^{T}$ Mb.

To straightforwardly demonstrate and analyze the energy consumption of a UAV flying with the two trajectories, the instantaneous power and total energy consumed are plotted against time in Figure 10. Two serrated blue curves denote the propulsion power of the HC and FCC trajectory designs. The minimum value of two power curves indicate that the drone is in straight-level flight, which are the same values under the two designs. As the forward speed of the quadrotor UAV was chosen as $V_{minE}$ in Figure 3, the maximum travel distance in linear flight is achieved. The maximum value of the two power curves represent the power of circular flight in the FCC design and hover flight in the HC design. Note that the proplusion power of hover flight is 0.915 kW, while the proplusion power of circular flight is 0.845 kW, which is less than that of hover flight. This simulation result is consistent with the previous conclusion. In addition, the acceleration and deceleration times are very short and, thus, can be ignored [15], leading to the discontinuities in the propulsion power curves between two flight states in both the HC trajectory and the proposed FCC trajectory.

As for the energy consumption of completing the total mission in Figure 10, the red dotted line represents the variation in UAV energy consumption under the HC design. The red solid line denotes the variation in UAV energy consumption with time under the proposed FCC design. Note that the energy consumption of the UAV seems to be a straight line, which means a constant propulsion power. However, it is actually composed of lines segment by segment which have different slopes, as linear flight and circular flight have distinct propulsion powers. The numerical results show that the energy consumption of the proposed FCC trajectory is 1918 kJ. As a comparison, the energy consumption of the HC trajectory is 1959 kJ, which means 2% energy is saved for 10 IoT users. It is worth noting that the energy consumption is also related to the distribution of IoT users, the data amount of IoT users to be collected and the altitude of the UAV. The energy-saving percentage is approximately between 1–10% when selecting an appropriate altitude, appropriate amount of data and random distribution of IoT user locations. In fact, the FCC design does save more energy than the HC design, as the energy consumption of circular flight is lower than that of hover flight. For further verification, the influence of IoT user’s data amount and UAV’s altitude on UAV’s energy consumption are presented in Figure 11 and Figure 12, respectively.

In Figure 11, we chose the amount of data to be collected from 10 IoT users as $Q_{data}={\left[1.2,1,0.1,0.7,0.4,1.6,0.7,1.1,0.4,1.3\right]}^{T}$ Mb when the ratio of data amount is 1. The blue line represents the variation in propulsion energy for HC trajectory when the amount of data increases from 1 to 10 times. The red line denotes the variation in propulsion energy for FCC trajectory. When the amount of data increases and other parameters remain unchanged, the optimal radius and velocity of the circular trajectory remain the same as the previous simulation results. In addition, the propulsion energy consumption of UAV increases by nearly 7 times due to the longer communication time. Moreover, the gap between the energy consumption in the two designs is more obvious with the increase in data amount.

In Figure 12, the variation in propulsion energy with the change in UAV’s altitude is plotted. The blue line represents the variation in the UAV’s energy consumption when the altitude of the UAV switches from 100 m to 200 m under the HC trajectory. The red line denotes the variation in the UAV’s energy consumption under the proposed FCC trajectory. As mentioned in the previous section, the optimal radius and speed of the circular trajectory are also determined by the altitude of the UAV. When the altitude of the UAV changes from 100 m to 200 m, the optimal radius and speed of the circular trajectory to calculate the energy consumption are changed, correspondingly. The gap between the energy consumption under the two designs is more evident with the increase in altitude, as the proportion of the circular trajectory’s radius in the communication distance becomes smaller. It can be concluded that the FCC design is more suitable for larger amounts of communication data and higher altitudes than the HC design.

## 5. Conclusions

This work studied energy-optimized UAV communication via trajectory planning based on a novel energy-consumption model, comprehensively considering UAV dynamics, BLDC motor dynamics, and aerodynamics. Due to aerodynamic effects, the propulsion power does not increase proportionally with the forward speed. For a linear trajectory, the UAV speed that minimizes the propulsion power is not the optimal speed to achieve the minimum energy consumption, as it needs a longer time to travel a fixed distance, resulting in a higher energy consumption. Based on these conclusions, for an IoT-user data collecting mission, a UAV may save more energy if it is circling around an IoT user instead of simply hovering above an IoT user. In the end, an energy-optimized trajectory is proposed by designing the circular trajectory and applying a TSP algorithm. Numerical results show that the proposed algorithm achieves less energy consumption than that based on a coarse TSP trajectory in a UAV-assisted communication network which requires large amounts of communication data and a high-altitude UAV.

## Figures and Tables

**Figure 1 sensors-22-08729-f001:**
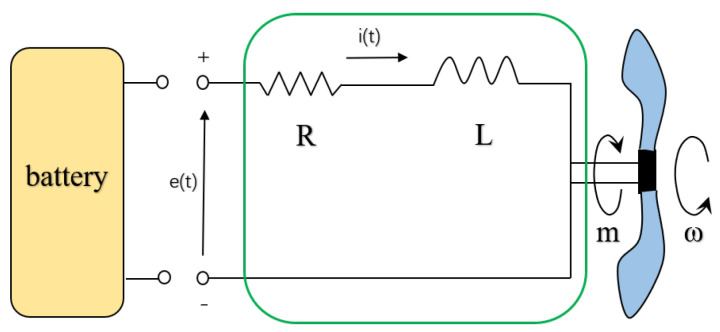
The BLDC electrical motor model for a rotary—wing UAV.

**Figure 2 sensors-22-08729-f002:**
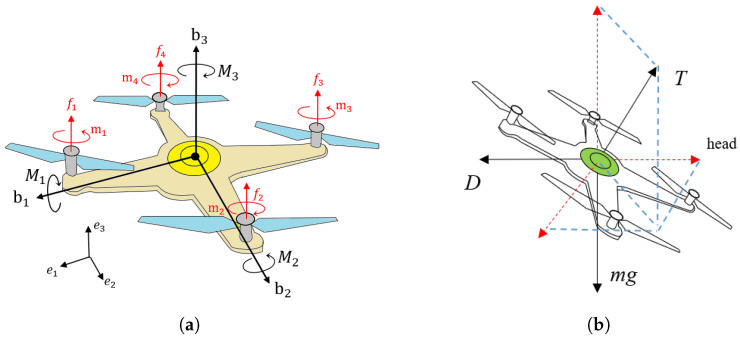
(**a**) The force and coordinate-system analysis. {e} represents the inertial reference frame, {b} stands for the rigid body frame, $m_{i}$ and $f_{i}$ are thrusts and torques produced by blades. (**b**) The external force analysis of the UAV.

**Figure 3 sensors-22-08729-f003:**
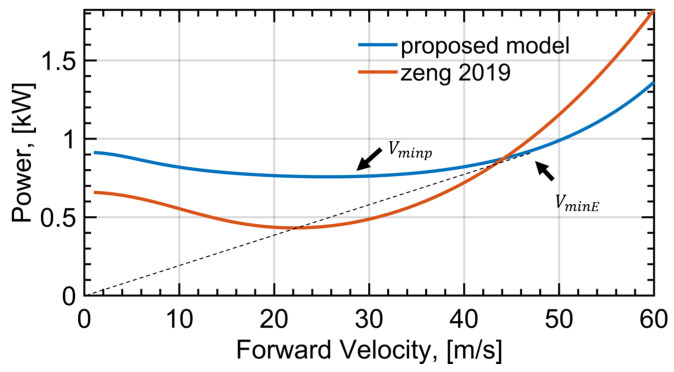
The forward speed *V* versus propulsion power *p*. Following an increase in *V*, *p* decreases firstly and then increases due to aerodynamic effects. The blue curve represents the proposed model and the red curve represents the model in [15].

**Figure 4 sensors-22-08729-f004:**
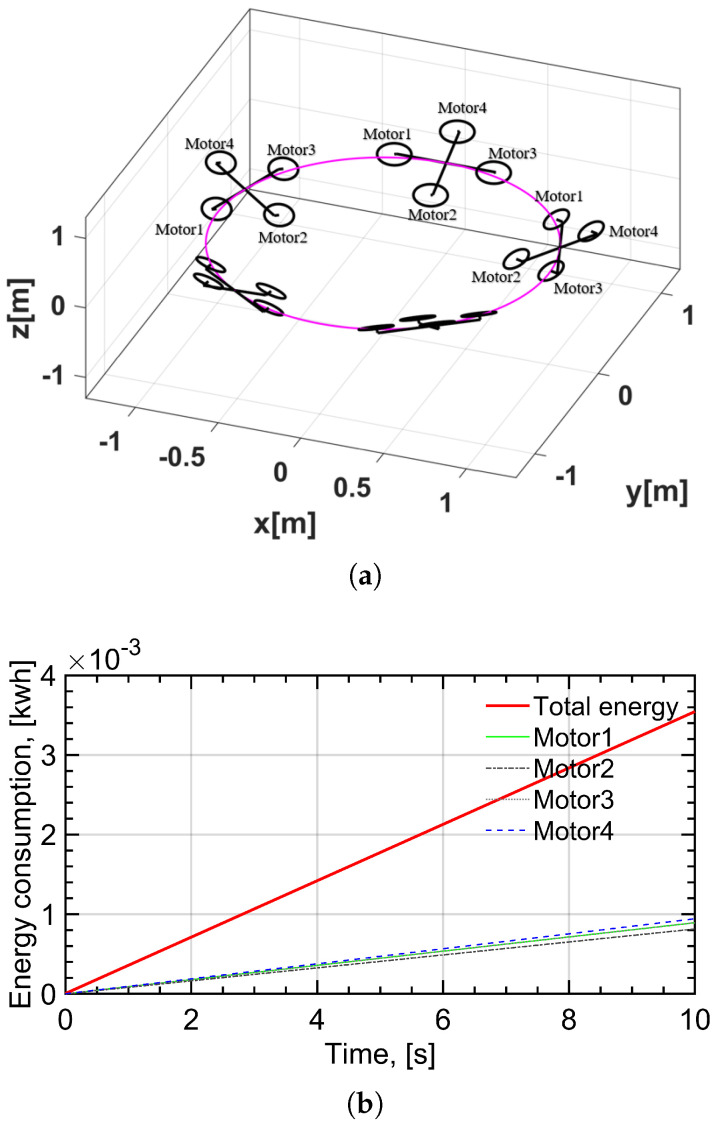
(**a**) An intuitive display of the circular trajectory in horizontal plane. (**b**) Total and single energy consumption of BLDC motors under circular trajectory. The red solid line represents the total energy consumption under circular trajectory. Motor1, Motor2, Motor3 and Motor4 represent the energy consumption of four motors, respectively.

**Figure 5 sensors-22-08729-f005:**
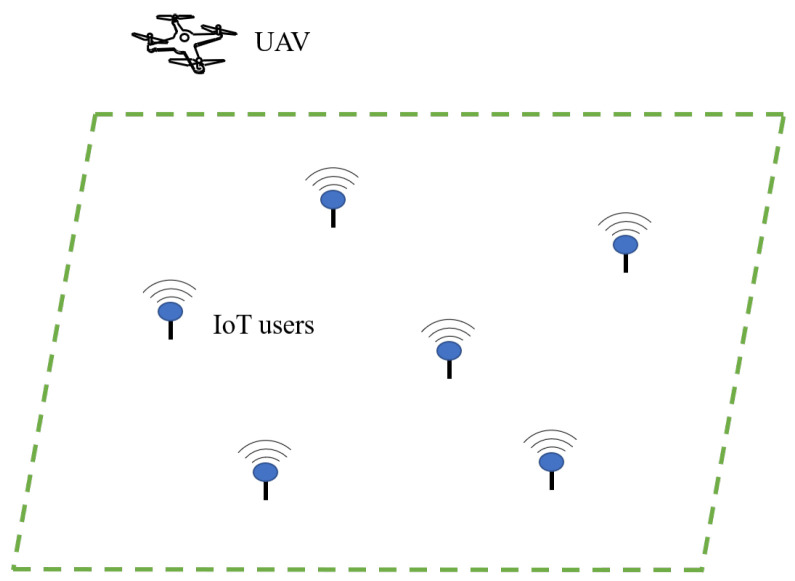
A quadrotor UAV is employed to collect data from a group of *K* IoT users in a UAV-assisted wireless communication network.

**Figure 6 sensors-22-08729-f006:**
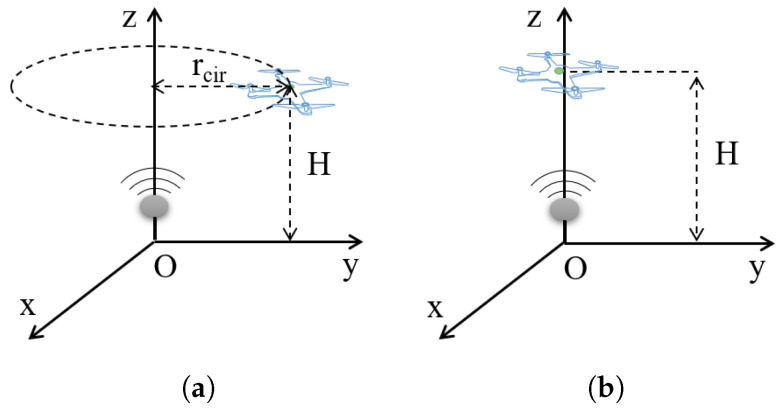
Two flight trajectories are comparied in the UAV-assisted wireless communication network. The vertical distance between the IoT user and the UAV is *H*. (**a**) The circular flight. (**b**) The hover flight.

**Figure 7 sensors-22-08729-f007:**
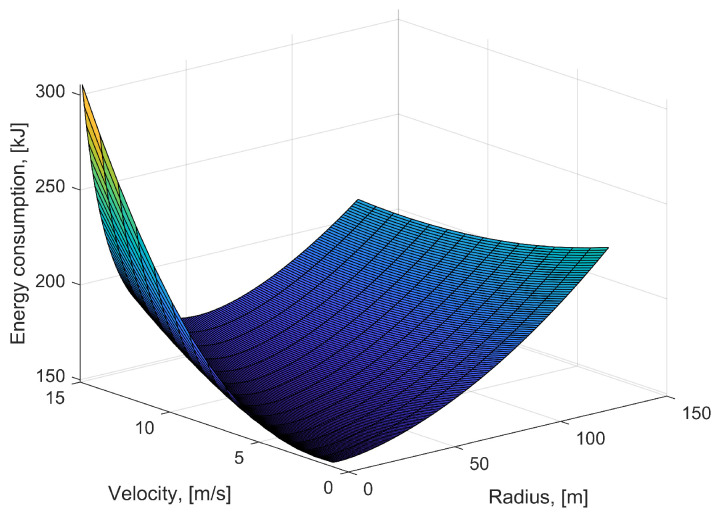
3D graph of energy consumption of circular flight with respect to radius and flight velocity.

**Figure 8 sensors-22-08729-f008:**
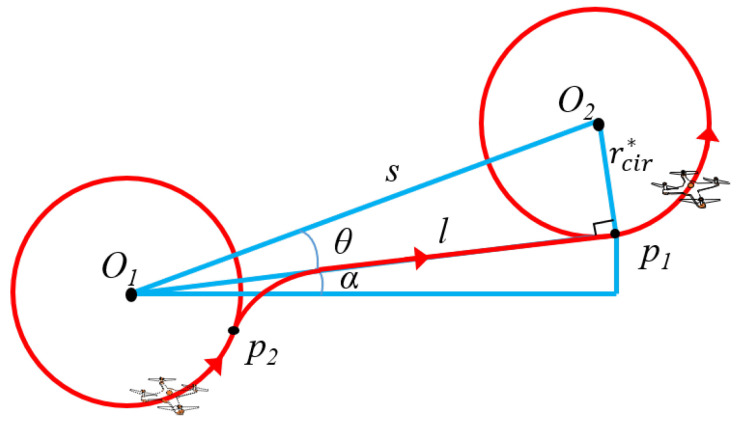
The calculation of the entry position of a Dubins arc.

**Figure 9 sensors-22-08729-f009:**
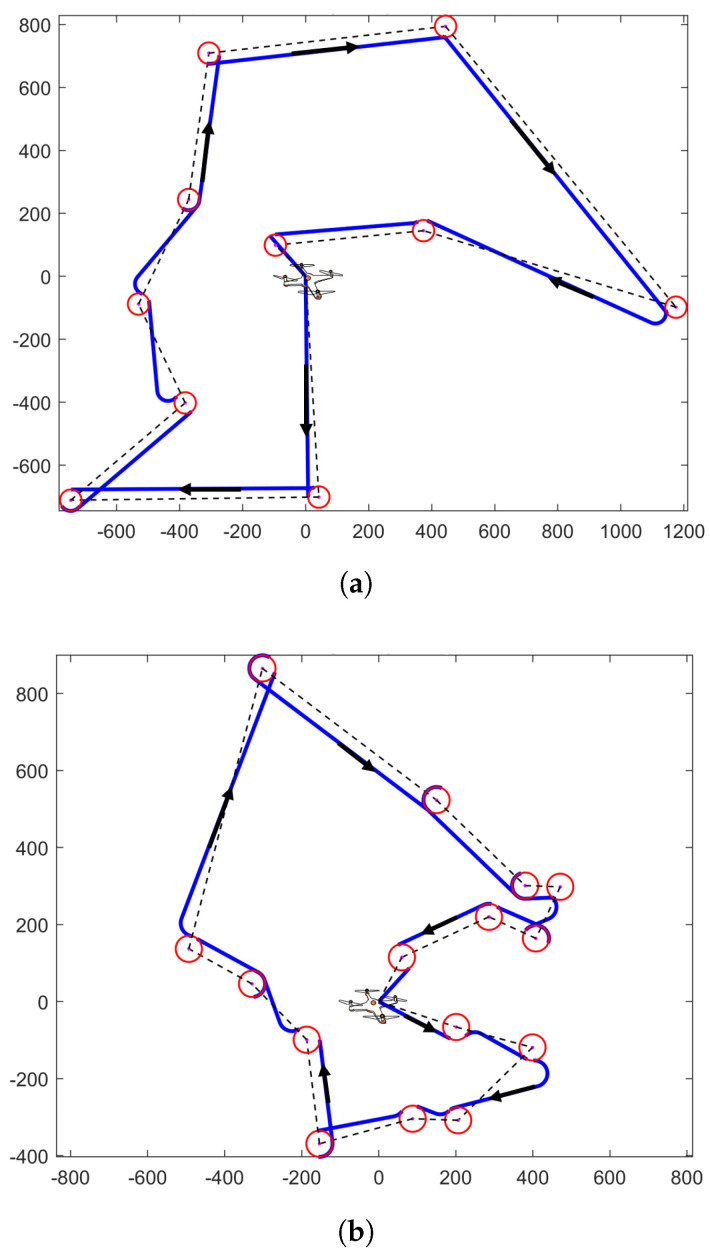
HC and FCC trajectory design in a UAV—assisted communication network. The positions of the circles’ centers represent IoT users. HC trajectory is denoted as black dotted line, which is formed by connecting the positions of IoT users in the order solved by TSP algorithm. (**a**) The proposed FCC trajectory for 10 IoT users. (**b**) The proposed FCC trajectory for 15 IoT users.

**Figure 10 sensors-22-08729-f010:**
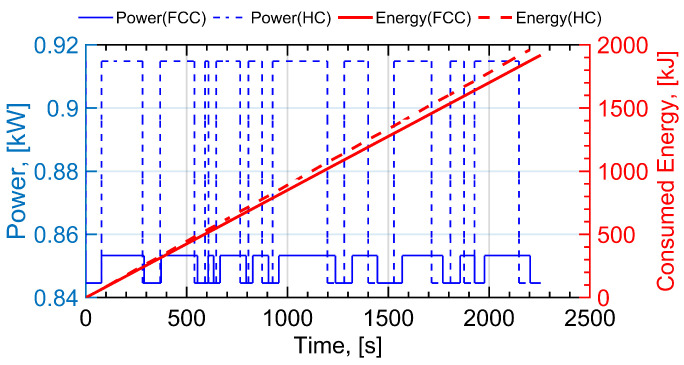
The variation in propulsion power and energy for 10 IoT users is plotted when the altitude of UAV is 200. The serrated blue curves denote the propulsion power. The red dotted line represents the variation in UAV energy consumption with time under HC design. The red solid line denotes the variation in UAV energy consumption with time under the proposed FCC design.

**Figure 11 sensors-22-08729-f011:**
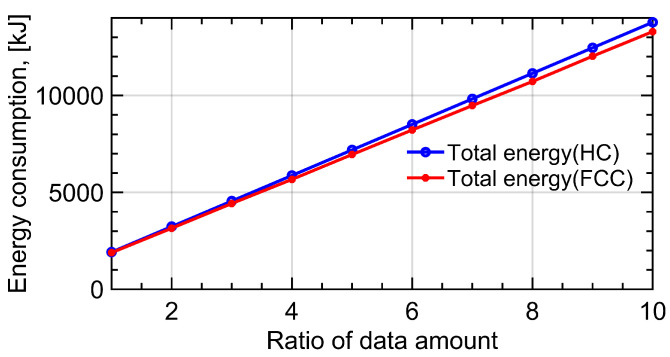
The variation in propulsion energy with the change in IoT user’s data amount. The blue line represents the variation in propulsion energy for HC trajectory when the amount of data increases from 1 to 10 times. The red line denotes variation in propulsion energy for FCC trajectory.

**Figure 12 sensors-22-08729-f012:**
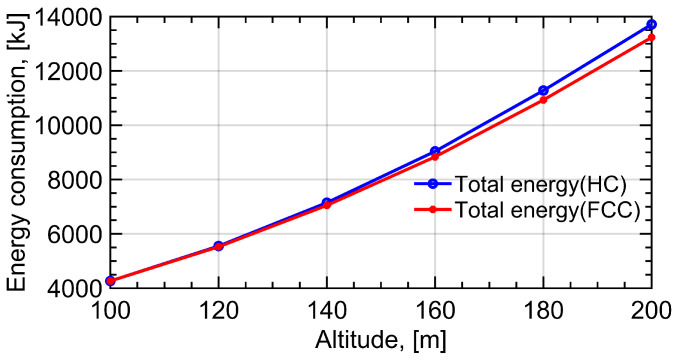
The variation in propulsion energy with the change in UAV’s altitude. The blue line represents the variation in UAV’s energy consumption when the altitude of UAV switchs from 100 m to 200 m under HC trajectory. The red line denotes the variation in UAV’s energy consumption under the proposed FCC trajectory.

**Table 1 sensors-22-08729-t001:** Model parameters of a quadrotor UAV [16,22].

Parameter	Value	Parameter	Value
mass	m=1.3 kg	gravity	g=9.8 N/kg
rotor radius	r=0.12 m	rotor location	l=0.4 m
lift slope	a=5.7	rotor disc area	A=0.0452 m${}^{2}$
fuselage equivalent flat plate area	$S_{F}=0.003$ m${}^{2}$	air density	ρ=1.225 kg/m${}^{3}$
collective pitch angle	$\theta_{0}=0.13$ rad	profile drag coefficient	δ=0.012
incremental correction factor	k=0.1	rotor solid	s=0.05
viscous damping coefficient	$D_{f}=2\times 10^{-4}$ Nms/rad	voltage constant	$K_{E}=0.01$ Vs/rad
motor resistance	R=0.2 Ω	moment of inertia x	$I_{x}=0.082$ kgm${}^{2}$
moment of inertia y	$I_{y}=0.084$ kgm${}^{2}$	moment of inertia z	$I_{z}=0.137$ kgm${}^{2}$

**Table 2 sensors-22-08729-t002:** Communication model parameters.

Parameter	Value
altitude of UAV	H=200 m
data amount	Q(i)=randn(1,30) Mb
transmission power	p=5 w
white-noise power	σ=−110 dBm
channel bandwidth	B=60 MHz

## Data Availability

The data that support the findings of this study are available from the corresponding author.

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
