# Peer review of "Energy-Efficient Trajectory Planning for Smart Sensing in IoT Networks Using Quadrotor UAVs"

_sensors, 2022, doi:10.3390/s22228729_

Round 1

Reviewer 1 Report

In this manuscript, an interesting framework had been presented for evaluating the optimal trajectory path of UAVs used to collect data from IoT-based sensors in the field, while attempting to minimize the energy consumption of the UAV during its round-trip flight. A detailed model was developed for calculating the UAV's power consumption travelling on a certain trajectory, and this mathematical model was further used to determine the optimal trajectory for minimizing these power consumptions. Overall, the results from the numerical examples have demonstrated the merits of the proposed framework which would be of great interest to any that are seeking to optimize the UAV's operations in such applications. 

The one minor confusion in the article is in the study given in Section 4.4 which was presented as an analysis aimed at "validating" the numerical results. It is not clear how such an analysis can validate the results, since the results appear to be generated from mathematical models that capture the actual performance characteristics under certain assumptions. What appears to be shown instead in that section is a comparison of the results from the proposed model and that from the HC model, which was good at highlighting the merits of the proposed scheme. A validation of the results could instead be accomplished by comparing the results returned from the model and one that is measured from an actual implementation of the same solution in the field. 

Author Response

Thank you for your careful reading and insightful comments. The original word “validate” is not appropriate and we feel sorry to misinform the readers. This paper focuses on the algorithm of trajectory planning considering UAV’s dynamics, which is mostly ignored by the community. Then we compare the proposed algorithm with relevant algorithms that have been verified in the literature. The verification/validation of the accuracy of the energy model is indeed significant and could be another focus of the current research in the future. An experimental validation could be conducted in the future, which we believe falls slightly out of the scope for this community.

Necessary wording is corrected and highlighted on page 14, line 360-362.

Reviewer 2 Report

1. The authors claim "smart sensing" while I didn't find explicit description on it. This concept should be explain in depth.

2. The dynamic model and energy consumption model seem to be similar with existing works. The difference or innovaiton should be highlighted.

3. The proposed trajectory planning seems strongly depend on IoT user distribution, which might not be energy efficient. More analysis should be put on this part.

4. The section II on energy model are too loose. The reviewer strongly suggest to shrink this part and extend the section III.

5. Pls explain the eq.(10), especially the three parts' meaning and how to specify them in your case.

6. More related literatures are encourage to include as follows:

(1) "3D UAV Trajectory Design and Frequency Band Allocation for Energy-Efficient and Fair Communication: A Deep Reinforcement Learning Approach," in IEEE Transactions on Wireless Communications

(2) "Trajectory Design and Access Control for Air–Ground Coordinated Communications System With Multiagent Deep Reinforcement Learning," in IEEE Internet of Things Journal

(3) "Resource Scheduling Based on Deep Reinforcement Learning in UAV Assisted Emergency Communication Networks," in IEEE Transactions on Communications

7. Simulation results in Fig. 10 are difficult to understand. How is the 10% gain achieved?

Author Response

  1. The authors claim “smart sensing” while I didn’t find explicit description on it. This concept should be explained in depth.

Reply: Thank you for your careful reading and insightful comments. Some literature claim that smart sensing techniques include UAVs, intelligent sensors and wireless sensor networks. To be specific, smart sensors are integrated with visual and mobile monitoring systems that are assisted by UAVs.  We have used this concept and corresponding changes are added and highlighted in the introduction on page 1, line 22-31.

  1. The dynamic model and energy consumption model seem to be similar with existing works. The difference or innovation should be highlighted.

Reply: Thank you for your careful reading. The proposed energy consumption model is derived from fundamental BLDC motor equations and UAV dynamic equations, comprehensively considering the influence of aerodynamics on torque and thrust coefficient. A prevailing (Zeng, yong et. al) energy consumption model does not consider the issue of BLDC motor efficiency, while another popular (Morbidi, F. et. al.) model does not consider the influence of aerodynamic effects. The differences are highlighted on page 2, line 63-67, and page 3, line 113-114.

  1. The proposed trajectory planning seems strongly depend on IoT user distribution, which might not be energy efficient. More analysis should be put on this part.

Reply: Thank you for your careful reading and insightful comments. Indeed, various applications such as one UAV to multiple static IoT users, multiple UAVs to multiple static IoT users and one UAV to multiple dynamical IoT users are out there to be studied. This work focuses on this first kind and proposes trajectory planning on known IoT user distribution. Additionally, as mentioned in the end of “Application Studies”, the proposed trajectory planning is suitable for the condition of large communication data and high altitude of UAVs. Numerical examples have demonstrated the merits of the proposed trajectory planning in such applications. Corresponding analyses are added and highlighted on page 13, line 356-358, page 15, line 410-411 and page 16, line 433-436.

  1. The section II on energy model are too loose. The reviewer strongly suggest to shrink this part and extend the section III.

Reply: Thank you for your constructive comments to refine our work. This paper has focused on the energy consumption model based on fundamental BLDC motor dynamic equations. In order to derive the model, BLDC motor dynamics and quadrotor dynamics are required, which take up a lot of pages. Yes, the energy model in Section 2 is indeed too loose. Hence, the derivation of the equations is reduced and corresponding changes are highlighted in Section 2 and Section 3. In addition, the equations of aerodynamics are placed in the Appendix to realize the consistency of derivation.

  1. Pls explain the eq.(10), especially the three parts’ meaning and how to specify them in your case.

Reply: Thank you for your careful reading. The first term on the right-hand side of (10) means that an acceleration is generated by the resultant external force acting on the UAV. The second term is the gravitational force in the opposite direction of e3. The third term is the fuselage drag, which is opposite to the forward direction. Corresponding changes are highlighted below eq.(10) on page 5, line 149-151.

  1. More related literatures are encouraged to include as follows:

(1) “3D UAV Trajectory Design and Frequency Band Allocation for Energy-Efficient and Fair Communication: A Deep Reinforcement Learning Approach,” in IEEE Transactions on Wireless Communications

(2) "Trajectory Design and Access Control for Air–Ground Coordinated Communications System With Multiagent Deep Reinforcement Learning," in IEEE Internet of Things Journal

(3) "Resource Scheduling Based on Deep Reinforcement Learning in UAV Assisted Emergency Communication Networks," in IEEE Transactions on Communications

Reply: Thank you for your suggestion to refine our work. The state of the art is now improved and corresponding changes are highlighted in the introduction on page 2, line 47-49 and line 73-74 ([17,18,29]).

  1. Simulation results in Fig. 10 are difficult to understand. How is the 10% gain achieved?

Reply: Thank you for your helpful comment. The left axis of Fig.10 represents power consumption and corresponding data are drawn with blue lines. The right axis represents energy (integral of power) consumption and corresponding data are drawn with red lines. The range of gain percentage is obtained through a large number of simulations with various IoT user distributions and amount of date to be collected in MATLAB environment. The results showing approximately 1%-10% less energy consumed by a UAV under versatile applications.

Round 2

Reviewer 2 Report

I dont have  more concerns.